BMDD: a novel approach for IoT platform (broker-less and microservice architecture, decentralized identity, and dynamic transmission messages)

Nguyen Lam Tran Thanh 1 thanhlam.bk.2110@gmail.com
Ha Son Xuan 2
Le Trieu Hai 3
Luong Huong Hoang 4
Vo Khanh Hong 4
Nguyen Khoi Huynh Tuan 4
Nguyen Anh The 4
Dao Tuan Anh 4
http://orcid.org/0000-0001-8800-4919 Nguyen Hy Vuong Khang 4
1 Department of Telecommunications Engineering, Ho Chi Minh City University of Technology , Ho Chi Minh , Vietnam
2 University of Insubria , Varese , Italy
3 Cantho University of Technology , Can Tho , Vietnam
4 FPT University , Can Tho , Vietnam
Al-Hadhrami Tawfik
Electronic publication date: 2022 Apr 22
Publication date: 2022
Volume: 8
Electronic Location ID: e950
Received 2021 Nov 2; Accepted 2022 Mar 25
Copyright: © 2022 Nguyen et al.
Copyright year: 2022
Copyright holder: Nguyen et al.
License: This is an open access article distributed under the terms of the Creative Commons Attribution License, which permits unrestricted use, distribution, reproduction and adaptation in any medium and for any purpose provided that it is properly attributed. For attribution, the original author(s), title, publication source (PeerJ Computer Science) and either DOI or URL of the article must be cited.
License URL: https://creativecommons.org/licenses/by/4.0/

Keywords: Internet of Things, Broker-less architecture, Microservice architecture, Decentralized identity, Dynamic message, IoT platform, gRPC protocol, Role-based access control, Kafka message queue

Funding: FPT University, Vietnam This work was supported by the FPT University, Vietnam. The funders had no role in study design, data collection and analysis, decision to publish, or preparation of the manuscript.

==============================
Undeniably, Internet of Things (IoT) devices are gradually getting better over time; and IoT-based systems play a significant role in our lives. The pervasiveness of the new essential service models is expanding, and includes self-driving cars, smart homes, smart cities, as well as promoting the development of some traditional fields such as agriculture, healthcare, and transportation; the development of IoT devices has not shown any sign of cooling down. On the one hand, several studies are coming up with many scenarios for IoT platforms, but some critical issues related to performance, speed, power consumption, availability, security, and scalability are not yet fully resolved. On the other hand, IoT devices are manufactured and developed by different organizations and individuals; hence, there is no unified standard (uniformity of IoT devices), i.e., sending and receiving messages among them and between them and the upper layer (e.g., edge devices). To address these issues, this paper proposes an IoT Platform called BMDD (Broker-less and Microservice architecture, Decentralized identity, and Dynamic transmission messages) that has a combination of two architectural models, including broker-less and microservices, with cutting-edge technologies such as decentralized identity and dynamic message transmission. The main contributions of this article are five-fold, including: (i) proposing broker-less and microservice for the IoT platform which can reduce single failure point of brokering architecture, easy to scale out and improve failover; (ii) providing a decentralized authentication mechanism which is suitable for IoT devices attribute (i.e., mobility, distributed); (iii) applying the Role-Based Access Control (RBAC) model for the authorization process; (iv) exploiting the gRPC protocol combined with the Kafka message queue enhances transmission rates, transmission reliability, and reduces power consumption in comparison with MQTT protocol; and (v) developing a dynamic message transmission mechanism that helps users communicate with any device, regardless of the manufacturer, since it provides very high homogeneity.

Introduction

In recent times, science and technology have developed significantly, which leads to stricter and more demanding needs from users. The current scientific landscape can be generally described as connecting the cutting-edge techniques as well as meeting the era’s demands. As the Fourth Industrial Revolution spread throughout the globe, all devices/sensors were able to connect to the wireless system (e.g., Bluetooth, Wifi, Zigbee) to increase the intra-interoperability among themselves and the different layers (e.g., devices – owners). In particular, the communication among the devices connect via a 5G/6G mobile network, the Internet for long-range transmission or wifi version 6.0, and Bluetooth version 5.0 for short-range connection. The need for increasing transmission speed has promoted research in the field of signal processing and wireless transmission. To cover that, all devices have internet capability, a.k.a Internet of Things (IoT). The released IoTs architecture platforms/models allows the user’s device connect to the Internet; thereby they are able to monitor and configure access control, enabling collecting and exchanging their data. This development is the main reasons for the birth of Big Data-related fields.

The Big Data data analysis challenge is not only a discrete and meaningless combination but helps providers perceive users’ behavior or habits to predict customers. As data becomes a new oil, new requirements for IoT platforms are set to ensure the privacy of users’ data (Ateş, Bostanci & Güzel, 2020). After obtaining data that has undergone analysis by Big Data, Machine Learning, and AI applications, devices can “learn” from the data itself and the owner’s feedback in specific cases; at this point, the concept of “smart devices” is introduced. IoT platforms also replace humans operating in dangerous places such as high mountains, deep oceans, or even outer space (Pillai et al., 2019). For Big Data or AI applications to operate effectively, a massive amount of data is sufficiently required to be rapidly and accurately collected, and the IoT platform is the main component responsible for this problem (Ulah et al., 2020).

IoT applications are increasingly diverse in many areas, including smart cities, healthcare, supply chains, industry, agriculture, etc. The number of IoT-connected devices worldwide is estimated to increase to 43 billion by 2023, three times more than in 2018 (Dahlqvist et al., 2019). Investments in IoT technology may grow at 13.6% per year through 2022 (Dahlqvist et al., 2019). However, in any application area, IoT platforms still share some of the most crucial aspects to evaluate an IoT platform: response time (27%), cost (18%), energy consumption (18%), reliability (14%), availability (14%), throughput (5%) and security (4%) (Asghari, Rahmani & Javadi, 2019). The above five aspects are interrelated, and depending on the application, the developer will focus on achieving a few specific features. Excluding the cost factor, in this article, we will evaluate the fundamental factors of the IoT platform. The following paragraphs first describe the five drawbacks of the current IoT architecture, then highlight our contribution for the new platform which focus on broker-less and microservice architecture, decentralized identity, and dynamic transmission messages (a.k.a BMDD IoT platform).

In terms of response time, every user wants to send or receive data as quickly as possible when they make a request. However, depending on the field of application, the response time of the IoT platform may be distinct (Hossein Motlagh et al., 2020). For example, when applying IoT to the medical field, the required response time should be the smallest as in emergency cases, the patient’s life sometimes depends on just a few seconds. On the other hand, for agricultural purposes, the response time can be flexible (https://www.emnify.com/blog/iot-connectivity-comparison-guide).

Some of IoT platform evaluation are energy consumption and reliability related to the protocol used in IoT platforms. Due to their small size, low performance, and limited processing ability, IoT devices often orient towards mobility (Pratap et al., 2019). Therefore, multiple protocols are introduced for IoT, such as MQTT, CoAP, HTTP, XMPP, AMQP, RESTful, Websocket, SMQTT, and DDS. Among them, MQTT is the most popular protocol and is used in both IBM, Microsoft, and Amazon IoT Framework (Fuentes Carranza & Fong, 2019). On the one hand, Bansal & Priya (2020) has evaluated the advantages and disadvantages of all nine protocols above and concluded that the DDS protocol satisfies the best response time and reliability criteria. Indeed, the MQTT protocol has three levels of QoS – 0, 1, and 2, equivalent to an increasing level of reliable transmission; the higher the QoS level, the greater the reliability of the transmission. On the other hand, to achieve this reliability, energy usage and response time have to trade-off. The QoS-0 level has the fastest transmission rate but the lowest reliability (Lee et al., 2013), while the QoS-2 level is the opposite. According to Toldinas et al. (2019), the power consumption of QoS-0 level is only about 50% compared to QoS-2 level, this means the higher the transmission reliability, the greater the power consumption while the transmission rate decreases.

Availability is another important aspect of IoT applications. There are two popular architectures: (i) architecture using a broker (brokering) represented by the MQTT protocol; (ii) the decentralized architecture represented by a machine-to-machine protocol, a.k.a broker-less (e.g., DDS). The most noticeable weakness of the first architecture is the single failure point (Trilles, González-Pérez & Huerta, 2020) since it requires a central broker to collect and coordinate messages, which affects the system scalability. On the other hand, the broker-less architecture allows devices to communicate directly with each other with extreme flexibility, high communication performance, and it is easy to expand the system (Bansal & Priya, 2020).

The security aspect is essential because the IoT platform is responsible for collecting data directly from user side. However, this issue has not received much attention as the current IoT devices have low processing capacity, and its protocols are not fully equipped with security mechanisms. The MQTT protocol has many drawbacks, namely authentication, authorization, and privacy mechanisms (Anthraper & Kotak, 2019). Nevertheless, there have been many articles proposing security enhancement mechanisms for MQTT (Nguyen et al., 2021; Thanh et al., 2021; Fremantle et al., 2014), they are mainly centralized forms of authentication that quickly lead to single failure points and are difficult to scale. Similarly, the DDS protocol also has inadequate security (Bansal & Priya, 2020). IoT systems, in general, are vulnerable to attacks due to user behavior. In particular, users often tend not to care about security issues, especially privacy, until damage occurs, such as loss of necessary data (Tawalbeh et al., 2020). For instance, the statistics of Subahi & Theodorakopoulos (2019) presented a significant percentage of IoT users are not aware of where they are sharing information.

In addition, uniformity (Di Martino et al., 2018; Noura, Atiquzzaman & Gaedke, 2019) is also a challenge for IoT platforms. Each IoT application collects and processes a different type of data; for instance, environment-related applications collect only one float value related to temperature or humidity, while coordinate-related applications collect a group of three values, including longitude, latitude, and altitude. This point leads to building multiple different applications even though the main feature is data collection. Besides, there are a significant number of service providers and manufacturers of IoT devices, leading to inconsistencies in data collection and transmission.

From the above analysis, the paper’s innovations and contributions are as follows:

Proposing a broker-less and microservices IoT Platform architecture – BMDD: broker-less architecture eliminates the reliance on a central broker for data collection and distribution of control commands. Brokering architecture is very easy to cause a single failure point when there are many users and devices connected to the IoT Platform simultaneously, which inevitably every IoT Platform has to encounter. In addition, the broker-less architecture also allows services to operate independently of each other, meaning that a service failure will not affect the entire operation of the system. Besides, BMDD also follows a modern microservice design that allows scaling out of the system efficiently and grants permission for BMDD to act as an open platform for third-party applications to integrate easily. The problem of monitoring and fixing errors is also an advantage of systems that follow the microservice design.

Proposing a decentralized authentication method with a strict model in the management of users, things, and channels: BMDD uses decentralized identifier (DID) technology based on blockchain technology for users and IoT devices. The nature of objects participating in the IoT network is highly decentralized; therefore, an IoT platform that allows relatively decentralized authentication will have significant relevance and allow the system to be scaled easily. Furthermore, authentication using DID technology allows to take advantage of the benefits of blockchain, and the device’s DID token remains forever compared to Single Sign-On and Oauth protocol authentication systems where these tokens always have an expiration time and have to request a new token after a period, which is quite inconvenient. Moreover, DID allows rapid authentication, only about 0.9 s–1.1 s, which is entirely suitable for actual deployment and is easily revoked via the API. In addition, BMDD provides user management service according to RBAC architecture and user organization model according to model tree, allowing to quickly isolate one or a group of users when the system is attacked. In addition, BMDD also provides APIs for device and channel management.

Proposing to apply gRPC protocol and dynamic message feature to ensure high homogeneity: The gRPC protocol is applied instead of the popular MQTT protocol because it is suitable for building a broker-less architecture. In addition, gRPC is also appropriate for all kinds of IoT applications (clarified in the background section) and provides excellent performance (Thanh et al., 2021b; Thanh et al., 2021a). Most importantly, the gRPC protocol provides the ability to build dynamic messaging, which allows IoT applications to be undiscriminating when collecting different data types such as string, float, integer, or JSON, contributing to the uniformity of IoT platform deployments for different domains without the need to reprogram services, also does not concern IoT device manufacturers.

The rest of the paper is organized as follows: Section 2: Existing research and related work.

Section 3: Technologies used in the paper.

Section 4: Detailed description of IoT Platform BMDD.

Section 5: IoT platform BMDD deployment.

Section 6: Test scenarios setting, the evaluation, and discussion.

Section 7: Conclusion and future development direction.

Related work

Broker-less systems

Industrial internet of things

The Industrial Internet of Things (IIoT) is a network of sense, management, and communication devices collaborating to attain common goals while satisfying the industrial environment’s requirements (Panda et al., 2020). To increase the safety and effectiveness of industrial processes, IIoT integrates several technical types of equipment (e.g., sensors, communication equipment) (Tran, Yu & Riedl, 2020). Also, IIoT is commonly used to collect, process, and transmit data from industrial environments for both centralized (Lewandowski et al., 2020) and decentralized (Luchian et al., 2021) architectures.

Research in Quirós, Cao & Canedo (2020) designed a dynamic reprogramming about IIoT-applied devices based on distributed automation. This paper allowed the allocation of computational tasks on the channel based on each node’s specific role and context. Some Coaty solutions’ applications for collaborative IIoT were presented in the studies of Seitz et al. (2018a, 2018b). These modules introduce augmented reality (AR) and blockchain-based application markets in an industrial circumstance. Edge devices were categorized by their available resources, the apps and the services they could access. Blockchain acts as a technology that allows application traceability validation to ensure transparency for the whole system. Chenaru et al. (2015) debated the integration of wireless sensor networks with cloud systems as back-end infrastructure supporting complex monitoring. Meanwhile, the fog computing model have been used to exploit on-node computing resources to efficiently utilize the restricted radio communication channels (Mihai et al., 2018). The work in Stamatescu, Stamatescu & Popescu (2017) presented the consensus algorithms that might represent a possible solution for distributed computing. Localization models could be built on the gathered data for online inference in numerous process summaries such as energy management (Stamatescu et al., 2019).

These proposed approach provided a real-time, deterministic, and re-programmable network behavior that the system used to leverage device capabilities and to schedule the computation workloads in the network while providing real-time guarantees. The novel runtime systems also made them possible to execute dis-persible code in existing off-the-shelf devices leveraging currently evolving technologies.

However, in the above approaches, the connections between the actors were loosely coupled, and it did not need an entity to manage the communication among them. This potential affects the entire system if a malicious user might take over a device and led to a collusion attack (Yassein et al., 2017). For instance, IoT devices are very vulnerable to malicious users or hardware devices such as node detection, corruption, eavesdropping, denial of service attacks, and routing attacks such as sinkholes and wormholes (Fang et al., 2020). Therefore, a decentralized identity mechanism is essential in the current IIoT environment.

Cyber-physical systems

Edward Lee proposed cyber-physical systems (CPS) to link virtual space and physical reality via the combination of networks, computers, and storage in National Science Foundation 2006 (Lee, 2006). In particular, CPS is an automated distribution system that integrated physical devices with a communication network based on a computer infrastructure platform (Wang, Törngren & Onori, 2015). The main focus of this protocol is on networking different devices in an IoT environment (Hermann, Pentek & Otto, 2016). It, therefore, includes a control unit that can regulate sensors and actuators, connect with the physical world, process the data obtained and swap them with other systems and/or cloud services by means of the communication interface (der Technikwissenschaften, 2011).

An important feature of CPS is its ability to obtain information and services in real-time (Boyes et al., 2018). Hence, CPSs have been applied to several areas, such as Healthcare (Liu et al., 2018), Smart Home (Shih et al., 2016), Transportation (Möller & Vakilzadian, 2016), and at most popular in Computers’ Network (Khan et al., 2016; Zanni, 2015). In other words, CPS could be thought of as a system that sent and received data from devices over a single network (Bagheri et al., 2015).

However, in addition to the listed advantages, CPS needs to consider its stability, trustworthiness, effectiveness, and security in operations (Cheng et al., 2016). With this in mind, one of the top priorities of CPS is to accommodate a high level of security supporting in all layers of the CPS architecture, guarding private data while also providing acute anonymity of data (Alguliyev, Imamverdiyev & Sukhostat, 2018). Besides, for the reliability of the CSP, the requests to send and receive data are also very important, where very small vulnerabilities can have huge consequences (Thanh et al., 2021c).

Microservice

The microservices architecture addressed the traditional monolithic issues; for instance, when designing a new application is a unit consisting of many components. The application is reasonably divided into modules, and each module serves a function. The monolithic approach shows its limits as the number of functions increases and the app gets larger and larger. Indeed, in a short time, developers cannot fully understand the entire system, which leads to serious problems during construction and maintenance. Therefore, microservices architecture was introduced to fill this gap (Bixio et al., 2020). This section focuses on summarizing prior work that used the microservices benefits in designing their architecture rather than detailing how it works and the direction of development.

Microservice framework

Amazon offers AWS IoT Greengrass (OSGi Event Admin Service, https://docs.osgi.org/specification/osgi.cmpn/7.0.0/service.event.html) as a solution to migrate analytics capabilities directly to edge devices (e.g., smartphones). Essentially, it is a program that permits running AWS Lambda features locally once installed on an edge gadget. AWS Lambda allows the device to run code without provisioning or controlling servers. Microsoft Azure provides similar functionality to Azure Stream Analytics on IoT Edge (Maven, https://maven.apache.org/), enabling users to use near-real-time analytic functionalities by using Azure Stream Analytics on IoT-applied devices. Besides, Micro-Mu (https://micro.mu/) is an open-source microservices toolset, giving a full stack for building and managing small services. Micro-Mu supports APIs to query microservices used as gateways for external access using Transport Layer Security (TLS). Azure Stream Analytics is a complex event processing and real-time analytics engine. Like AWS IoT Greengrass, the working principle allows edge devices to locally perform Azure Stream Analytics rules. In addition, Google Cloud IoT (Ops4j pax url, https://ops4j1.jira.com/wiki/spaces/paxurl/overview) integrated the Apache Beam SDK (Senseioty, https://flairbit.com/senseioty-iiot-solution/), providing a rich set of session and windowing analysis primitives. It presented a united advancement pattern for defining and implementing data processing pipelines across various stream handling engines, including Apache Flink, Apache Samza, Apache Spark, etc.

For the limitation of these platforms, AWS IoT Greengrass neither provide any functionality to move Lambda calculation to and fro between cloud edge devices nor mix with exterior stream processing engines. Again, a limitation of the Azure Stream Analytics tool is that there is no mechanism for dynamically converting rules between the edge and the cloud. For the Apache Beam, this platform only supported core/cloud-appropriate extensible engines and it is not designed to support edge analytics.

Microservice application

Maia and associates (Simeoni et al., 2021) presented IRRISENS, which is designed based on microservices architectures used in agricultural environments to sense soil, crop, and atmospheric parameters. IRRISENS interacted with third-party cloud services to schedule irrigation and could control automatic irrigation. For their results, the proposed framework could add new devices (IoT) to the existing system. It also was possible to successfully expose the interaction model to clients’ applications. Moreover, security metadata was added to the description, stating what is needed for secure access to a thing. In this case, the Security schema (JSON Web Token Bearer) to be used in order to access to specific content that exposes how to access and interact with the Smart House resources. In addition, Pratama et al. (2019) introduced a smart collar based on microservices backend design for dairy cow behavior monitoring. The platform’s backend is created using the Microservices architecture idea, where each progress is isolated into services independent from one to another. Each service has specific tasks conveyed through the Rest API. This platform supports five services, namely Auth, Sensor, Monitoring, Learning, Front-end services. To a greater extent, Badii et al. (2019) have implemented a specific set of Microservices for IoT Applications in the Smart City context. Specifically, they carried out a Microservices set for Node-RED, enabling the formation of a great variety of new IoT applications for smart cities, including dashboards, IoT Devices, data analytics, discovery, and the corresponding Lifecycle. It has enabled the creation of a wide range of innovative IoT Applications for Smart Cities, involving Dashboards, IoT Devices, GDPR-compliant personal storage, geographical routing, geo-utilities, data analytics. The proposed solution has been validated against a large number of IoT Applications for the cities of Firenze, Antwerp and Helsinki. Therefore, to exploit these benefits, we apply the microservice architecture to design our proposed framework in IoT platform.

Security and privacy issues

IoT device vendors desired to develop smart IoT devices based on their benefits, which had led to heterogeneous device production and possible cross-platform conflicts; meanwhile, data generated from IoT devices should be kept safe and not misused (Gheisari, Wang & Chen, 2020). These two issues created significant challenges in providing a trusted environment in the IoT platform. It was beyond the scope of this paper to provide a detailed assessment of data security requirements. Instead, in related work, we focused on the currently popular approaches, including the access control models and data encryption, and provided their pros and cons.

Encrypted data approaches

Research in Di Crescenzo et al. (2013) proposed PICADOR, a secure topic-based pub/sub-system through the proxy re-encryption scheme. Brokers re-encrypted the publications, the authorized subscribers then restore these publications’ plaintext in their systems. However, this re-encoding increases the computational cost dramatically at the broker point. The subject matter of each publication is shipped off the agent in plaintext, which could not fulfil the confidentiality assurance conditions of the participants.

A high-level study project on Content Centralized Networks (called CCNx or Named Data Network – NDN) based on the pub/sub system was introduced in Saadallah, Lahmadi & Festor (2012). CCNx uses a hierarchical naming approach (Jacobson et al., 2009) to organize data content. Unlike the traditional MQTT system relying on end-to-end delivery, CCNx exploits a broadcast-based to convey information rather than broker-based. However, this issue might not apply to malicious publishers; for instance, if a malicious publisher uses the fake labels or re-uses another publisher’s public key, the subscribers in CCNx may get the faulty publication.

To address this issue, Trinity (Ramachandran, Wright & Krishnamachari, 2018) was introduced as a blockchain-based distributed pub/sub broker. Specifically, Trinity checks the immutability of transactions in the same system by integrating the broker system in the same blockchain network. Trinity also addresses the ordering problems by using a blockchain-based system. However, authentication problems can seriously affect the security of the entire system, since a malicious device (e.g., malicious code, controlled by an attacker) could replace one benign device of the users that might seriously affect the security of the whole system (Tariq et al., 2010).

Lu et al. (2017) introduced a resolution to gather data while conserving the confidentiality of fog computing-enhanced IoT devices. Authors used a one-way hash string to authenticate IoT devices, then applied the Chinese Remainder Theorem to aggregate data generated by different IoT devices. They also took advantage of Uniform Paired Encryption to provide data security. However, their resolution applied a common privacy rule, differential privacy, to all IoT devices causing unintentional amounts of sensitive data to be accepted. Applying the access control model can solve this drawback, where only authorized objects can accessed data (Son, Dang & Massacci, 2017; Xuan et al., 2016).

Besides, Anusree & Sreedhar (2015) proposed a secure broker-less sub/pub system. In particular, this approach was proposed to use the Elliptic curve-based identification mark encoding algorithm; thereby, they effectively reduce the cost of encryption and decryption on the respective parties. In other words, thanks to the Elliptic curve-based encryption algorithm, the costs related to computation and communication costs no longer affect the whole system. However, the key privacy issues still existed and had not been addressed in such a mechanism (Thanh et al., 2021; Nguyen et al., 2021). Furthermore, complete encryption of the data generated by IoT devices is not strictly necessary since not all data is sensitive, depending on the usage context; for example, privacy-preserving in a healthcare environment (Duong-Trung et al., 2020a, 2020b) and cash-on-delivery (Ha et al., 2020a, 2020b).

In this article, to balance the above problems, we only apply Blockchain to build a DID model for authentication purposes. To protect the IoT data, we apply a decentralized mechanism based on a role-based access control model instead of encrypting all data.

Access control models

Sunghyuck (Hong, 2020) demonstrated that security threats could occur in each component of the IoT. Therefore, IoT devices and their generated data needed an access control method to protect the device from unauthorized personal device intrusion (Shi et al., 2015). Jain, Kesswani & Agarwal (2020) introduced a service delivery and utility manager (SDUM) as an intermediate data access layer between data owners and data consumers. The data consumer sends a command to SDUM for accessing the data using a unique personal ID. Then data access floor transits the user’s verification order to KMaaS (i.e., key management as a service). A one-time password (OTP) is utilized to check a correct user prior to sending information; afterwards data access layer confirms the user with an OTP code. SDUM transfer scrambled data to the corresponding user via a safe communication route. Besides, Rashid et al. (2020) proposed Enhanced Role-Based Access Control (ERBAC) for securing IoT medical data in Health Care Systems in terms of its storage on public cloud systems. The model was implemented by using the Model View Controller Framework (MVC) of Microsoft (Smith, 2018).

However, the bouncing model only uses basic authentication methods based on the IDs of the data owners and data consumers. This point is not suitable for IoT devices as well as problems with multi-level users where the lower level received the permissions to access data from the upper ones (Thi et al., 2017). Moreover, instead of dealing with security requirements such as collecting data from IoT devices, defining roles for each data user, this paper focuses on data management in the Microsoft’s Azure cloud. These above solutions did not solve the key problems of access control in the IoT platform; it merely introduced the traditional solution for data protection based on the RBAC model (Son & Chen, 2019).

Background

This section addresses details regarding the technologies used to build the BMDD. Besides, selected technologies and current popular technologies are also compared to support the article logically.

Internet of things protocol

IIoT devices are small and possess low processing capacity; thus, the protocol selection for the communication process is crucial, determining the architecture of the entire IoT Platform, taking into account these key factors: transmission speed, reliability, power consumption, scalability, and security.

MQTT protocol

MQTT (Message Queuing Telemetry Transport) is a small network protocol operating under the publish-subscribe mechanism according to ISO and OASIS standards (https://mqtt.org/). The main concepts of MQTT include MQTT broker, MQTT client (publisher and subscriber), MQTT topic.

MQTT Broker is the junction of all incoming connections from the client and is the center of systems that uses the MQTT protocol. The principal responsibility of the Broker is to receive messages from publishers, queue them, and then forward them to subscribers based on topics.

MQTT client is classified into two groups: publishers and subscribers, in which the prior is the message sender to the Broker and the latter is the message receiver from the Broker. A client can act as both a publisher and a subscriber on one or more specific topics.

The topic is a logical concept that MQTT uses to route messages sent from publisher to subscriber; for example, when the publisher sends a message to topic A, only topic A subscribers can receive the message. The model has the advantage that every user who subscribes to the topic receives a message from the publisher. However, sensitive data is considered an enormous weakness in terms of security.

Another concept of MQTT is Quality of Service (QoS). The MQTT protocol has three QoS levels including QoS-0, QoS-1, and QoS-2 which are defined as follows (Archana et al., 2020): QoS-0 (maximum one time): one packet is transfered to a destination up to once.

QoS-1 (minimum one time): each packet is emitted to the destination at least once, expecting packet iteration to occur.

QoS-2 (precisely one time): each packet is shipped to its target sole one time.

The transmission process of MQTT according to each QoS level (https://www.hivemq.com/blog/mqtt-essentials-part-6-mqtt-quality-of-service-levels/) is shown as in Fig. 1:

Figure 1 MQTT QoS level communication.

Figure 1 indicates that transmitting a message with the highest accuracy (QoS-2) using the MQTT protocol requires at least four steps, which reduce the transmission speed, increase the number of connections as well as increase the energy usage. Although the article (Thanh et al., 2021) proposed a model combination between MQTT and message queue to make use of the quick transmission rate, the tiny energy consumption of QoS-0, as well as ensure the reliability of QoS-2, however, there is still a weakness that the publisher does not receive any notification to ensure the message reception. In addition, when the subscribers receive the messages, the MQTT protocol also does not support sorting them in order (Soni & Makwana, 2017), which can seriously affect the accuracy of the data.

In addition, MQTT has another noticeable concept, “keep-alive” (https://www.hivemq.com/blog/mqtt-essentials-part-10-alive-client-take-over/), which is a function of the MQTT broker that allows maintaining the connection so that communication is available between the broker and the client. When the keep-alive time exceeds, and the client does not send any messages to the broker, the broker will disconnect from the client. At this point, the client must send a “PINGREQ” packet to notify the broker that it is still operating, requesting to reconnect, then the client can send the message to the broker. In a practical scenario with thousands of clients, the broker has to handle many PINGREQ packets that can overload the broker. The keep-alive feature allows optional maintenance of a constant connection between the client and the broker to reduce energy consumption. However, this feature also has the weakness of not ensuring system availability, especially in emergencies.

gRPC protocol

gRPC (general-purpose Remote Procedure Calls) protocol (https://grpc.io/) is an open-source system of superior effectiveness of the Remote Procedure Call (RPC) protocol improved by Google in 2016. The gRPC is constructed on top of the http/2 protocol, which contains multiple advancements over earlier forms to permit greater and more productive HTTP connections. The gRPC provides machine-to-machine (M2M) communication. One of the most important aspects of http/2 is multiplexing, allowing us to deliver and gain many packages in a sole connection in comparison with http/1.1 as illustrated in Fig. 2. This feature is one of the main advantages that make us choose gRPC as it allows the client to open a single connection and transmit a collection of different data to the server, which saves energy during data transmission. In addition, gRPC carries a wide variety of programming languages and is perfectly suitable for embedded devices (Karcher et al., 2020).

Figure 2 Overview of multiplexing in http/2 (Du, Lee & Kim, 2018).

The gRPC protocol provides for four types of machine-to-machine communication (Indrasiri & Kuruppu, 2020) as in Fig. 3:

Unary: This type is similar to conventional client-server communication. The client sends a request to the server; afterwards, the server processes and transfers the outcome to the client.

Server streaming: In this type, the server receives a request from the client, after processing, sends back a stream of data to the client. The client will be notified when they reach the end of the data stream. The sequence of messages of every stream is ensured to be identical between client and server.

Client streaming: Like server streaming, in this type, the client will be the side sending the data stream to the server. After reading and operating the required processes, the server delivers the response to the client.

Bi-directional streaming: In this approach, data will be sent from both directions, client and server, independent from each other. Also, the client and server can deal with the data stream independently, meaning that when the client delivers information to the server, the server can conduct a particular task (while still accepting the messages coming from the client) and transfer the response back to the client (while the client is still conveying other messages).

Figure 3 M2M communication in gRPC (Indrasiri & Kuruppu, 2020).

There are some features that MQTT does not provide. Regarding the first benefit compared to the MQTT protocol, gRPC offers four types of machine-to-machine communication, which make it highly appreciated due to its suitability with the diversity of IoT applications. For example, when applying IoT in agriculture, the unary communication type is suitable to collect temperature data, similar to the ability provided by MQTT. However, considering the application of IoT for the medical field with many sensors working simultaneously, and each sensor may stream a bunch of data in a different period, the client streaming communication type is a better option than MQTT. Apart from that, with the ambition of building BMDD towards adaptation with many areas of IoT application, gRPC is the optimal choice since it provides a variety of communication forms. Moreover, a prominent advantage of gRPC over MQTT is that it guarantees message ordering in streaming data cases; both the sender and the receiver get notifications when the transmission is successful, which is probably a superiority.

Dynamic message

The gRPC framework provides a special data type, “Any” (https://developers.google.com/protocol-buffers/docs/proto3#any). This data type allows the client and server to freely exchange messages regardless of the data form. The “Any” type also supports the four types of communication methods that gRPC provides. This feature is handy in constructing BMDD because it provides the ability to transfer many different types of data without developing a single type of service, increasing the compatibility of BMDD. Therefore, it can be applied to many different fields using IoT technology, contributing to the increment of the homogeneity of systems. For instance, a customer of a health care service running on the BMDD platform often uses transmission devices provided by company A. When this customer travels to another country, but the hotel only has B carrier equipment, the incompatibility between devices is negligible when using BMDD. The communication process between the client and the server using the “Any” data type is summarized as follows:

Step 1: build a function to send data (function putData) on the client side with the data type “Any”:

func putData(valueType string, value []byte)

error {

  conn, err := grpc.Dial(address, grpc.WithInsecure())

  if err != nil {

   log.Fatalf("did not connect: %v", err)

  }

  defer conn.Close()

  client := proto.NewDataServiceClient(conn)

  ctx, cancel := context.WithTimeout(context.Background(), time.Second*3)

  defer cancel()

  reqData := &proto.PutDataRequest{

   Data: &any.Any{Value: value},

   Type: valueType,

  }

  resp, err := client.PutData(ctx, reqData)

  if err != nil {

   log.Printf("put data error. %v" err)

   return err

  }

  if resp.Err != 0 {

   log.Printf("put data exec error. %v", resp.Desc)

   return err

  }

  return nil

}

Step 2: When sending data, the client attaches the type of this data. This is illustrated in function testPutData()

func testPutData() {

  putData("int", []byte("1"))

  putData("float", []byte("1.1"))

  putData("bool", []byte("true"))

  putData("string", []byte("this is a string msg"))

  putData("json", []byte("{\"msg\":\"this is a json msg\", \"other\":1}"))

}

Step 3: After receiving data in form of bytes from the clients, server will base on the data type to parse the byte data with the correct data type (function parseAnyData())

func parseAnyData(valueType string, anyData *anypb.Any) (interface{}, error) {

  if anyData == nil {

   return nil, errors.New("illegal param")

  }

  var err error

  var reqData interface{}

  switch valueType {

  case "int":

   reqData, err = strconv.ParseInt(string(anyData.Value), 10, 64)

  case "float": 

   reqData, err = strconv.ParseFloat(string(anyData.Value), 10)

  case "json":

   err = json.Unmarshal(anyData.Value, &reqData)

  case "bool":

   reqData, err = strconv.ParseBool(string(anyData.Value))

  case "string":

   reqData = string(anyData.Value)

  default:

   reqData = string(anyData.Value)

  }

  return reqData, err

}

We also provide the source code via the GitHub links (https://github.com/thanhlam2110/dynamic-message-client.git; https://github.com/thanhlam2110/dynamic-message-server.git).

Kafka message queue

Kafka (https://kafka.apache.org/) is a dispersed texting scheme, able to transmit a huge number of messages in real-time, where the information will be deposited on the message line and on the disk to guarantee security if the receiver has not received it.

Kafka architecture consists of four principal elements: producer, consumer, topic, and partition. The producer element is the client to distribute messages into topics. Data transmitted to the topic’s partition is stored on the broker. Kafka consumers can be multiple clients who subscribe to and receive messages from similar or different topics classified by group names. Data is conveyed in Kafka by many topics which can be created when passing data to varied applications is necessary. For the BMDD platform, the Kafka topic concept is equivalent to a transmission channel created and managed by each user. The Partition’s function is to store the topic data, and and each topic can at least have partitions. Each Partition can store data permanently and assigns them an ID called offset. Moreover, a set of Kafka servers, known as brokers and zookeepers, is a broker management service.

Because gRPC does not support message caching, and to increase the availability of BMDD, we add a Kafka message queue. All messages exchanged between services in BMDD must go through Kafka. This method allows the service to be recovered when it fails, and messages will not be lost during downtime due to its storage on Kafka for a configurable amount of time.

Microservice and monolithic architecture

Monolithic architecture is an architecture software in which different components (such as authorization, business logic, notification module, etc.) are combined into a single program from a single platform (Gos & Zabierowski, 2020). Microservice is an approach to develop an application using a set of small services, each of which runs its process independently, usually an HTTP resource API. This modern architecture allows more extensive, more complex, and scalable applications (Tapia et al., 2020).

To clarify the monolithic and microservice architectures, the description of the image below illustrates the application of these two architectures in building an E-commerce application, as in Figs. 4 and 5.

Figure 4 Monolithic flow on the example of e-commerce application (Gos & Zabierowski, 2020).

Figure 5 Microservice flow on the example of e-commerce application (Gos & Zabierowski, 2020).

Monolithic architecture’s advantage is its ease to be developed and deployed. However, the complicated maintenance mechanism, low reliability (one failure can crash the entire application), availability (must be re-deployed for update), and low scalability (Gos & Zabierowski, 2020) are the drawbacks of this structure.

The downside of monolithic architecture is the benefit of microservice architecture and the applications of which are simple to maintain since the modules are completely separate from each other. It also provides high scalability and high reliability as a service failure does not affect other services. The only weakness of the microservice architecture is the complex deployment (Gos & Zabierowski, 2020).

This paper prioritises constructing BMDD under microservice architecture because of the advantages it provides, simplifying the expansion and new features integration processes for BMDD in the future.

Decentralized identity

DID (Decentralized identifiers) is a novel way to identify data objects which DID’s controller specifies, designed to perform the identification separating from a third party identifier. With the utilization of URIs, DID establishes the association between a subject with a document to build up reliable interactions, allowing control of the verification without testifying with a third party. Each DID material can reveal cryptographic material, confirmation techniques, or services, enabling trusted interactions to subject and prove control by DID controller.

Sidetree

Sidetree is a protocol that allows users to init, manage the global and unique PKI metadata controlled by their user. Thanks to the connection to the existing decentralized system such as Bitcoin, Ethereum, …, Sidetree protocol also has open, public, and permissionless and other properties.

Figure 6 elucidates the structure of Sidetree-based DID overlay network:

Existing decentralized networks, such as Bitcoin, Ethereum, e.g., will be connected to SideTree via DID operations. The operations of DID consist of four main functions: creating, updating, recovering, and deactivating, responsible for querying, initializing, and updating data in the decentralized system.

Sidetree protocol is organized into nodes located between the decentralized system and the CAS network, interacting with the decentralized network via operations according to predefined rules.

CAS (Content-addressable storage) stores and distributes operation files having an obligation to define and store DID operations. Expressly, the Core Index File declare the create, recover and deactivate operators; meanwhile, Provisional Index File contains operator updates. Besides, Core Proof File and Provisional Proof File contain necessary cryptographic proofs of operators like signatures, hashes, etc. Finally, SideTree uses COMPRESSIONALGORITHM to generate a Chunk Files file which contains the operation source data.

Figure 6 Sidetree-based DID overall network.

Figure 7 elucidates the file structures of the side tree protocol: The protocol comprises of three file constructions, which keep DID administration information and are meant to support principal functionality to facilitate light node configurations, minimize forever the reserved data, and guarantee resolution effectiveness of DID.

Figure 7 File structures of Sidetree protocol.

DID URI Composition: The SideTree protocol uses a unique identifier segment to identify a DID method called DID Suffix; when a Sidetree-based DID is generated, the unique identifier corresponding to it is also generated. These DID URIs are used to identify a corresponding method correctly and can be self-certifying. DID URI can be presented in two primary forms: short-form and long-form.

Short-form URI: Short-form URI: The SideTree protocol uses a hashing algorithm to generate a Short-Form DID URI from JSONCANONICALIZATIONSCHEME; the short-form of the DID URI has the format:

did:METHOD:<did-suffix>

Long-form URI: Long-form URI: The formation of Long-Form DID URIs is to prevent endless time situations between DID’s generation and anchor, propagate and process of DID operation by self-certifying and self-resolving capabilities. Structurally, Sidetree Long-Form DID URIs are the Short-Form DID URIs but appended with a new segment called long-form-suffix-data; the value of this segment is calculated from Operation Suffix, and Operation Delta then encoded via DATAENCODINGSCHEME function. Long-Form DID URI has format:

did:METHOD:<did-suffix>:<long-form-suffix-data>

JSON web signatures

Generally, the authentication mechanism of DID operations is implemented based on JSON Web Signatures and represented by two actions Signing and Verifying, consisting of four primary functions: create, recover, deactivate and update. Except for the create method, the others require key materials for verifying and authenticating the transactions. When a transaction is required to sign, that transaction must contain two parameters kid and alg.; while the alg is a required parameter and cannot be null, the kid can be declared and assigned a value or not. The format of the authentication string is:

{

 "kid": "did:example:example_kid_value",

 "alg": "example_alg_value"

}

The authentication process on DID is the same for each transaction if that transaction is required to perform Verification. In the first step, the kid is extracted and traversed to find an authentication method with the same id and kid pair. Then the found authentication method is converted to JWK to perform the JWS Verification process in the final step.

Proof of fee

The mechanism of the Sidetree network is designed to resist low-cost tampering, mainly for permissionless and open implementations, by leveraging public blockchains with two key characteristics: Base Fee Variable and Per-Operation Fee. Base Fee Variable is calculated from a set of variables belonging to a particular anchor system using defined functions. In addition, it can be used to set the lowest transaction cost corresponding to the number of DID operations calculate the costs of other economic setups effectively. Per-Operation Fee has the main purpose to validate the user’s transaction by performing the following processes sequentially: Determine the Base Fee Variable of the transaction or block.

Use Operation Count from Anchor String to Multiply the Base Fee Variable.

Validate the transaction anchored in the anchoring system.

Analysis and execute transaction if the transaction uses required fee.

The acronyms

We will use the acronyms which are shown in Table 1 for the whole article.

Table 1 The acronyms and their description.

Acronyms	Description	Acronyms	Description	
IoT	Internet of Things	RBAC	Role Based Access Control	
BMDD	Broker-less and Microservice	DID	Decentralized Identity	
	architecture, Decentralized identity, and	CSS	Control Service Server	
	Dynamic transmission messages	CSC	Control Service Client	
MQTT	Message Queuing Telemetry Transport	CDS	Collect Data Service Server	
gRPC	general-purpose Remote Procedure	CDC	Collect Data Service Client	
	Calls	LB	Load Balancing	
DBL	Database & Log	Org-1	Organization 1	
OMS	Object Management Service	Org-2	Organization 2	
DPS	Data Processing Service	User-A	User A	
RTM	Realtime Messaging Service	Dr-B	Doctor B	
MQ	Message Queue	Nu-C	Nurse C	
QoS	Quality of Service	CCU	Concurrent User	

Approach

Definition

In the background and related work section, we introduced the technologies used to build the BMDD platform and associated research directions. This section dedicates to user-related concepts, an essential subject of BMDD.

User organization

There are two main groups in BMDD: root users and normal users. The root user is the group that manages the BMDD platform, having full rights to the BMDD participants except for reading the user’s data. The group of normal users use the BMDD service, divided into two subgroups, parent and child users, supposing that the user group is an enterprise and wants to organize the user rights hierarchy. The child user connects to the parent user through the “user_parentid” attribute, which means that the child user’s “user_parentid” field is equivalent to the parent user’s “user_name” field. Every normal user is a child user of the root user. User information is stored in NoSQL format, allowing flexibility to add additional attributes for the user apart from the specified ones. The above user management model is called the model tree and is illustrated in Fig. 8.

Figure 8 User organization as model tree.

The parent user structure in JSON format is as follows:

{

  "userid":"76579b0e-e899-4fc9-a839-607c371b9585",

  "username":"organization1",

  "usermail":"representation@ organization1.vn",

  "userstatus":"ACTIVE",

  "userparentid":"root",

  "usertype":"representation"

}

And the child user structure in JSON format is as follows:

{

  "userid":"37389774-98a6-4dc2-a8b1-22fc29d321ba",

  "username":"thanhlam",

  "usermail":"thanhlam@organization1.vn",

  "userstatus":"ACTIVE",

  "userparentid":"organization1",

  "usertype":"normal"

}

The implemented model trees have two primary purposes. Firstly, BMDD can manage user hierarchy, suitable for enterprise deployment environments and individual customers. Secondly, the users have the right to build coherence and stricter management thanks to hierarchy management. For example, BMDD provides an API enabling the parent user to disable all child users through the status field. When the user status is “DISABLE”, the users and their devices will be isolated and cannot interact with the BMDD, which allows to quickly isolate a group of users when a hacker attacks the BMDD system from one user point. For the root user, this is to isolate an entire organization from the BMDD. Restoring can easily be done by changing the status to “ACTIVE”. The API also ensures that only the high-level user can change the low-level status being its child user.

User role

BMDD user role involves two concepts, including user role for the organization of users following the model tree (described in the user organization section) and role related to IoT service.

User roles related to IoT services include the following capabilities. First, the user has full authority to create logical information to manage his devices and communication channels. The communication channels are equivalent to the Kafka topic concept presented in the background. This information will be embedded into the device and used for checking the permissions when the device interacts with BMDD. This embedded process is beyond the scope of this article; thus will not discuss in detail. Besides, the user can assign device information to a communication channel, allowing users to strictly manage device and channel, only devices that have been authorized and assigned to a specific channel are allowed to send and receive messages on that channel. This approach also helps to reduce security risks caused by the careless behavior of users. Finally, the users also have control over sharing their devices with other users; for instance, a patient can give their doctor permission to access the device and receive heart rate data from a sensor, etc.

BMDD platform proposal

IoT Platform BMDD is designed according to broker-less and microservice architecture, including three main layers: thing layer, edge layer, and cloud layer, as described in the Fig. 9.

Figure 9 BMDD platform architecture.

Each layer include these services described as following:

The thing layer is the layer of physical devices (gateways) that receive information from sensors or users. Each gateway uses two services, including control service client (CSC) and collect data service client (CDC). The term “client” here distinguishes it from service of the same name implemented on the edge layer. Every gateway only needs these two services, regardless of application to any IoT field, due to the dynamic message feature that BMDD provides. In addition, CSC and CDC also operate independently following the microservice architecture. The role of the CSC is to receive control commands from the control service server (CSS) to control remote devices, while CDC has to send the collected data to the edge layer. In practice, the thing layer includes the gateway and IoT devices (sensor, actuator, etc.). In BMDD, the gateway is implemented on the user side and it communicates locally (LAN network) with IoT devices via wifi (Sajjad et al., 2020) or Bluetooth (Sathyaseelan et al., 2021) to collect data or control devices. For communication with the Edge layer (WAN network), the gateway use gRPC protocol via CDC and CSC service.

The edge layer includes three services: control service server (CSS), collect data service server (CDS), and load balancing (LB). CSS is responsible for receiving control commands from a valid user and then sending this command to the CSC. The role of the CDS is to receive data sent from the CDC, which is attached to the user’s DID token, thing-id, and channel-id. CDS initially sends this token to the decentralized identity service (DID) on the cloud layer for authentication, then sends thing-id and channel-id to the role-based access control service (RBAC) for permission check. If the data passes the authentication and authorization process, it will transfer to the message queue through CDS. LB acts as a proxy and handles the load distribution for the system.

The cloud layer includes service role-based access control (RBAC), decentralized identity (DID), message queue (MQ), object management (OMS), data processing (DPS), real-time messaging (RTM), and database-log (DBL). DID is responsible for authenticating the user with the DID token. The role of RBAC is to check user roles configured by the user to predefine permissions related to the data collection or device control process. RBAC is capable of providing APIs that allow users to build their access control policies flexibly. The implementation section will present the detailed scenario for the user role. MQ receives data from the CDS and control command from CSS and distributes it to other services. DPS receives data from MQ and performs advanced processing, e.g. threshold analysis. The result of the processing may be to determine the threshold levels that should be alerted to the user. RTM handles the process of sending notifications to users. Data processed by the DPS can be stored in the DBL. Lastly, OMS provides APIs that allow users to create/update/delete management information of devices and channels. It also allows upper-level users to create and manage lower-level users with a model tree.

Implementation

Overview

In the implementation section, we assume a BMDD-specific application scenario described as follows: two IoT service providers using the BMDD platform: smart home service provider (Org-1) and smart health service provider (Org-2). User-A, a customer of Org-1, decides to register to use Org-2’s remote health care service and is supervised by doctor B (Dr-B) and nurse C (Nu-C). This scenario frequently happens in practice, particularly during the Covid-19 pandemic.

User-A using smart home service with Gateway device can collect information about the house such as room temperature, humidity and can control home devices. Besides, the gateway also collects information related to heart rate, blood pressure, etc., as health care services are also used. User-A only allows the gateway to send its health-related data to Dr-B and Nu-C. At the same time, User-A only allows Nu-C to view health data while Dr-B has additional rights to control healthcare devices remotely.

To implement the above scenario through BMDD, User-A must perform three processes, including registering to use the service (1), creating management information for the devices and channels (2), and sending health data (3). Process 1, User-A registers to use IoT service to be granted DID token through DID service.

Process 2, User-A creates device management information, communication channels and then maps devices on these channels through the OMS service. Subsequently, User-A creates authorization roles through the RBAC service.

The authorization is specified as following: User-A has full access to his/her devices, including smart home devices and medical devices. User-A will assign permission only to allow the gateway to send health-related information to Org-B while smart home-related information is not allowed.

User-A authorizes dr-B to read medical data and control medical devices.

User-A authorizes nu-C to read medical data.

The third process is sending medical data of Org-1 from User-A to the medical staff of Org-2. This process is described in the Fig. 10, including steps from (1) to (9).

Figure 10 Overview model of data collection process through BMDD.

The steps of the data collection process are described as following: Step (1): The User-A body sensor sends health data to the gateway, stored locally.

Step (2): After a while, this data will be streamed by CDC to CDS with DID token of User-A.

Step (3): CDS sends User-A’s token to a decentralized identity (DID) for authentication. If the authentication process is successful, it can proceed next step (4); otherwise, all data will be dropped, and CDS will send an invalid token response to CDC.

Step (4): When the user’s token is authenticated, the CSC sends user_iD resolved from the user DID token to the RBAC service. The RBAC uses the user_iD to check user permissions. If permission is valid, it can continue to the next step (5). Otherwise, all data will be dropped, and CDS will send an invalid role response to CDC.

Step (5): After the authentication and permission checks, the health data will be sent to the MQ.

Step (6): DPS receives health data from MQ and conducts analysis.

Step (7): Data, after analysis, will be sent to RTM and evaluated according to a predefined threshold.

Step (8): Assuming the health data exceeds the allowable threshold, for example, User-A’s average heart rate in 5 min is 180, then RTM will send a warning message to MQ.

Step (9): CDC on the Org-B side receives a warning message from MQ and sends it to Dr-B and Nu-C.

Also, in this scenario, Dr-B uses only one process controlling the device. Nu-C has the right to view medical data, however for a brief presentation; we mainly present the procedures related to Dr-B as Dr-B also has additional control over the medical device. The appearance of Nu-C is to ensure that the modelling implementation scenario is more accessible to readers.

Dr-B will control emergency life support devices after receiving a warning that User-A’s heart rate exceeds the threshold. The process of controlling the device through BMDD is described in the Fig. 11, including steps from (1) to (6).

Figure 11 Overview model of the device control process through BMDD.

The steps of the device control process are described as following: Step (1): Dr-B sends control commands to CSC.

Step (2): CSC sends control command with Dr-B’s token to CSS.

Step (3): CSS sends Dr-B’s token to DID service for authentication. If the authentication process is successful, it can move to step (4): Otherwise, the control commands will be dropped, and the CSS will send an invalid token response back to the CSC.

Step (4): When the Dr-B’s token is authenticated, the CSS service sends Dr-B_iD resolved to the RBAC service from the Dr-B DID token. The RBAC will use Dr-B_iD to check Dr-B permissions. If permission is valid, it can proceed to step (5). Otherwise, the control commands will be dropped, and the CSS will send an invalid role response back to the CSC.

Step (5): After going through the authentication and permission check, the control command will be sent to the MQ.

Step (6): CSC on gateway of Org-1 side receives a control command from MQ and performs device control.

BMDD Platform in detail

The section above has shown the entire exchange process between services inside BMDD in a specific scenario. This section gives the details of each process. However, because BMDD is designed according to microservice architecture, the communication process of services is enormous and complicated, so we only focus on the most important processes, including: Requesting and resolving DID token.

Creating management information of devices, communication channels, and mapping devices to channels.

Authentication and authorization.

Collecting data.

Controlling device.

Detailed source code of BMDD can be found at GitHub link including decentralized identity service (https://github.com/thanhlam2110/bmdd-did-service), collect data service (https://github.com/thanhlam2110/bmdd-collection-service), role-based access control and object management service (https://github.com/thanhlam2110/bmdd-rbac-service).

Requesting and resolving DID token process

Through this process, the user can get the DID token. Before the user can send data and control the device, the DID token must be resolved to authenticate the user.

The process of requesting a DID token is presented in Fig. 12. The user sends his information to the DID service and receives the DID token in return.

Figure 12 Request DID token process.

The process of user registering has the following input and output in JSON format.

The input is

{

  "userName": "thanhlam",

  "serviceId": "vnptit",

  "serviceEndpoint": "vnptit"

}

And the out put is:

{

  "longFormURI": "did:ion:EiDQx4R6z…FRaWFdETXcifX0"

}

The resolving DID token process is as the Fig. 13 below. The user sends DID token information to DID service. DID service will resolve the DID token and return the user information if the authentication is successful.

Figure 13 Resolve DID token process.

The process of resolving DID token has the following input and output in JSON format.

The input is:

{

  "longFormURI": "did:ion:EiDQx4R6z…FRaWFdETXcifX0"

}

And the out put is:

{

  "didDocument": {

   "id": "did:ion:EiDQx4R6z…FRaWFdETXcifX0",

   "service": [

    {

     "id": "#vnptit",

     "type": "LinkedDomains",

     "serviceEndpoint": "vnptit"

    }

   ],

   "authentication": [

     "#thanhlam"

   ]

  }

}

Creating device, channels management information and mapping devices to channels process

Users can interact with OMS via this process to create management information of devices, channels and map devices to channels. This process helps users control their devices and communication channels. Users will clearly know where they share their data, which helps reduce security risk by user’s behaviors.

The process of creating Things management information is as the Fig. 14. The user sends information, including his DID token, along with device information to the OMS service. The OMS service will forward the user’s DID token to the DID service for authentication. If the authentication process is successful, OMS will generate Things info.

Figure 14 Create thing process.

The process of creating Things information has the following input and output in JSON format.

The input is

{

  "token":"user_did_token",

  "thing_name":"thing_name"

}

And the output is

{

  "thing_id":"thing_id",

  "thing_name":"thing_name",

  "thing_status":"thing_status",

  "assign_user":"user_id",

  "user_parent_id":"user_id"

}

OMS will automatically generate thing_id, thing_status values, where thing_id conforms to the UUID standard (https://www.ietf.org/rfc/rfc4122.txt) and is a unique value. The thing_status is used for determining the device state, whose default value is “ENABLE”. If the status is “ENABLE”, this means the Things will work properly; by contrast, it can not interact with the BMDD, if the status is “DISABLE”. The thing_status value can only be changed by the user that owns the Things. Moreover, the user_id value of the user who created the Things, obtained through the token resolution process, is equal to user_parent_id value. The “assign_user” attribute is also automatically generated with the initial value of user_id. When the users assign their device to some other users, this field will be added with the user_id of these assigned users, for example, when User-A assigns device control to Dr-B, the field assign_user has the value as an array “assign_user”: “[User_A_id, Dr_B_id]”.

The process of creating channel management information is as in the Fig. 15. The user sends information, including his DID token, and channel information to the OMS service through which the user’s DID token is transferred to the DID service for authentication. If the authentication is successful, OMS will create channel info.

Figure 15 Create channel process.

The process of creating channels information has the following input and output in JSON format.

The input is

{

  "token":"user_did_token",

  "channel_name":" channel_name"

}

And the output is

{

  "channel_name":"channel_name",

  "channel_id":"channel_id",

  "channel_status":"channel_status",

  "user_parent_id":"user_id"

}

OMS will automatically generate channel_id, channel_status values, where channel_id conforms to the UUID standard and is unique. The channel_status is used to determine the channel state, and the default value is “ENABLE”. If the status is “ENABLE”, the channel can transport data; by contrast, when the status is “DISABLE”, the channel can not transmit messages to BMDD. The channel_status value can only be changed by the user who owns the channel. In addition, the user_id value of the user who created the channel, obtained through the token resolution process, is equal to user_parent_id value.

The process of mapping the Things into the channel is as in the Fig. 16.

Figure 16 Map thing to channel process.

The user sends information to the OMS service, including his DID token and thing_id and channel_id information. The OMS service will forward the user’s DID token to the DID service for authentication. If the authentication is successful, OMS will map the Things information to the channel information.

The process of mapping Things information to channels has the following input and output in JSON format.

The input is

{

  "token":"user_did_token",

  "thing_id":"thing_id",

  "chanel_id":"chanel_id"

}

And the output is

{

  "map_id":"map_id",

  "thing_id":"thing_id",

  "chanel_id":"chanel_id",

  "user_parent_id":"user_id"

}

OMS will automatically generate map_id values, which conform to the UUID standard and are unique. The user_id value of the user who maps the Things to the channel, obtained through the token resolution process, is equal to user_parent_id value. Note that it is possible to map multiple things on a channel.

Authentication and authorization process

The DID service is responsible for authenticating users through the DID token. The RBAC is accountable for checking the roles that the user has pre-configured through the OMS service. DID and RBAC mainly support data collection and control device processes. Therefore, we will present this process combined with two processes of collecting data (interacting with CDS) and control service (interacting with CSS). The process of interaction is as in the Fig. 17. Before sending data or commands to MQ, CDS and CSS must undergo authentication and check-role processes. CDS/CSS sends the user’s DID token to DID authentication service; if successful, CDS/CSS will receive user_id information. Then, CDS/CSS sends information (user_id, thing_id, and channel_id) to the RBAC service to check if the user has permission to access the Things or the user has permission to the channel, or the Things is assigned to the channel. If the authentication and permission checking is successful, CDS/CSS can send data or control commands to the MQ. Otherwise, all data and control commands will be dropped.

Figure 17 Authentication and authorization process.

Data collection process

The data collection is the data-sending process from CDC to CDS, going through user authentication and permission checks. Finally, data will be sent to MQ, distributed to other services, depending on the application purpose. The interaction process is as the Fig. 18.

Figure 18 Collect data process in detail.

CDC streams data with the users DID token, thing_id, and channel_id to CDS. CDS service receives the data stream and interacts with the DID service and RBAC service to authenticate users and check roles. When the authentication and permission checking is successful, the CDS will stream the data stream to the MQ and return the status to the CDC. The data collection proceeds on the gRPC protocol.

Control device process

The control process sends control commands from CSC to CSS, going through user authentication and permission checking. Finally, the command will be sent to the MQ, distributed to the receiving CSC to control the device. The interaction process is as the Fig. 19.

Figure 19 Control device process.

The CSC sends control commands with the user’s DID token, thing_id, and channel_id to CSS. The CSS service receives control commands and interacts with the DID service and RBAC service to authenticate users and check roles. When the authentication and permission checking is successful, the CSS sends control commands to the MQ and returns the status to the CSC. The control device proceeds on the gRPC protocol.

Evaluation

After implementing BMDD, this section is for continuing to build scenarios to evaluate the performance, load capacity, and availability of BMDD. Due to the microservice architecture-based platform of BMDD, the services are deployed on Amazon EC2 virtual machines (VMs) in Singapore in the test scenario. Each service is equivalent to a VM with the following hardware and software configuration as Table 2.

Table 2 VM EC2 configuration.

CPU	1 vCPU	
RAM	1 GB	
Disk	8 GB SSD	
OS	Ubuntu 16.04	

The above configuration is a free EC2 server configuration provided by Amazon (https://aws.amazon.com/). There are many famous companies have deployed their service on Amazon EC2 (https://aws.amazon.com/ec2/customers/), so we chose EC2 to deploy the cloud layer. However, in this paper, we only use Amazon EC2 for implementing the proof-of-concept of the BMDD platform. Our purpose is to prove BMDD can be implemented, microservices can communicate with each other, and test performance. The deployments in production are heavily technical for example, using Docker (https://www.docker.com/) and Jenkins (https://www.jenkins.io/) pipeline for CI/CD deployment, Kubernetes (https://kubernetes.io/) for microservices management, ELK (https://www.elastic.co/what-is/elk-stack) for logging, etc., beyond the scope of the article. These parts are completely possible because we have provided the source code of BMDD.

In addition, Table 3 indicates the hardware configuration and the software of Raspberry Pi module B applied (https://www.raspberrypi.com/products/raspberry-pi-3-model-b/) to emulate the Gateway device.

Table 3 Configuration of Raspberry Pi module B.

CPU	Broadcom BCM2837, ARMv8 (64 bit) quad-core, 1.2 Ghz	
RAM	1 GB	
Disk	2 GB SSD	
OS	Raspbian	

We use the Raspberry Pi module because of its popularity and affordable price. Raspberry Pi module is also used for the gateway in many papers (Daidone, Carminati & Ferrari, 2021; Luchian et al., 2021; Pratama et al., 2019). Moreover, the Raspberry Pi has built-in Wifi and Bluetooth function which is suitable for local communication (LAN network) between gateway and IoT devices. However, as described in the BMDD platform proposal section, the gateway includes only two gRPC services (CDC and CSC) which can be implemented by many programming languages such as C/C++, Go, Python, etc. (https://grpc.io/); so we can use other microcontrollers for the gateway. Apache Jmeter software was used to simulate the number of concurrent users (CCUs) interacting with the BMDD during the evaluation. Apache Jmeter (https://jmeter.apache.org/) is an open-source, completely written in Java, utilised to test efficiency on both static assets, dynamic assets, and Web applications. It can mimic multiple virtual users, broad requests on a server, or a gathering of servers, networks, or objects to examine tolerance load tests or investigate response time. When using Jmeter to test the system load, at first, Jmeter creates requests and transmits them to the server in accordance with a predefined operation. After taking and collecting responses from the server, it presents the report data. Jmeter has numerous report factors but focuses on throughput and error when operating system load testing. Throughput (request/s) is the request quantities resolved by the server every second, and error (%) is the level of failed requests out of the total number of requests, wherein Error is defined as an overloaded service condition leading to the service being idle or unresponsive.

Scenario 1: The load test scenario of data collecting service

The first scenario aims to confirm the highest number of users and transmit data (Concurrent User – CCU) to the BMDD without the system error. This situation compares the performance between two architecturesystems, including broker-less systems using gRPC protocol (BMDD) and brokering systems using MQTT protocol. The test model is illustrated in Fig. 20.

Figure 20 Model of load test collect data process.

All edge and cloud layer services are deployed on the EC2 VM with the configuration shown in Table 2; only the things layer components (CDC and CSC) are deployed on the Raspberry Pi module as Fig. 21.

Figure 21 Gateway deployment on Raspberry Pi module.

We set up a data collection model at the things layer, data transmission at the edge layer, and data reception at the cloud layer. In this scenario, we assume the user is authenticated and has a valid role. We simulate the number of concurrent users streaming incoming messages using Jmeter software. Then, we gradually increased the number of CCUs and recorded the highest number of CCUs that did not generate errors for both brokering and broker-less architectures. The test results are shown in Figs. 22 and 23, respectively.

Figure 22 The collect data load test result: brokering architecture & MQTT protocol.

Figure 23 The collect data load test result: broker-less architecture & gRPC protocol.

The test results illustrated that the broker-less architecture using the gRPC protocol provides two times higher throughput than the brokering architecture using the MQTT protocol. In addition, the broker-less architecture also provides better load capacity (700 CCUs) compared to brokering architecture (500 CCUs).

Scenario 2: The speed test scenario of streaming data

The purpose of this scenario is to measure the message transfer rate. We do not change the number of users and use Jmeter to gradually increase the number of messages in one transmission and record the Round Trip Time (RTT) from when the message is sent at CDC to when it is received in MQ.

In this case, we also compare the transfer rate between broker-less architecture using gRPC protocol and brokering architecture using MQTT protocol. We also record the highest number of messages successfully delivered in a single communication without errors. Error is defined as the percentage of messages sent to the destination out of the total number of messages sent. Test model and service deployment environment at layers are similar to scenario 1. The test results are shown in Figs. 24 and 25, respectively.

Figure 24 The streaming data speed test results: brokering architecture & MQTT protocol.

Figure 25 The streaming data speed test results: broker-less architecture & gRPC protocol.

The test results illustrate that the broker-less architecture using the gRPC protocol provides powerful message streaming capabilities compared to the brokering architecture using the MQTT protocol. In addition, in this scenario, we also use a htop (https://htop.dev/) software that allows monitoring processes taking place in the system in real-time to evaluate CPU and RAM usage for two services CDC and CDS of broker-less architecture. The test results are shown in Table 4.

Table 4 CPU/RAM consumption when streaming data: broker-less architecture & gRPC protocol.

Messages		No load	10,000	50,000	100,000	
CDC	CPU	0	0	0.3%	0.7%	
(on Raspberry Pi)	RAM	0	0	2.5%	2.7%	
CDS	CPU	0	2.3%	8.2%	10.5%	
(on VM EC2)	RAM	1.7%	1.7%	1.9%	2.1%	

As a result, at no-load and 10,000 messages, we recorded a consumption level of 0, which means that the resource consumption of the services is very low (no-load) or happens for a brief time (10,000 messages), so the meter almost does not record the change in resource usage. The results show that the resource consumption is very low for the high load case (50,000–10,000 messages). The fact that Things is not constantly streaming large amounts of data like in this test shows that applying gRPC protocol to our BMDD gives very good results, suitable for devices with low hardware.

Scenario 3: The speed test scenario of authentication mechanism

This scenario compares user authentication speed between two authentication models, including centralized authentication using Single Sign-On (SSO) and decentralized authentication using decentralized identity (DID). The test model is illustrated as in Fig. 26.

Figure 26 Model speed test authentication process.

Both the Single Sign-On and the DID service are deployed on the EC2 VM with the configuration shown in Table 2. For the Single Sign-On service, we use open source CAS Apereo (https://www.apereo.org/projects/cas) and Oauth protocol. Both centralized and decentralized authentication models go through two steps. The first step is to request a token, and the user will use this token to authenticate. For centralized authentication, SSO will return the Oauth token, and for decentralized authentication, DID will return DID token. The second step is token validation process, also known as token resolve. In this test, we use Jmeter to simulate the number of concurrent users (CCUs) requesting tokens and then resolving the received tokens. We will record the highest CCU level without the system failing and two parameters throughput and error. The test results are shown in the table below. Centralized authentication model using SSO (Tables 5 and 6)

Decentralized authentication model using DID (Tables 7 and 8)

Table 5 The test request token results: centralized authentication using SSO.

Request token (CCU)	100	200	300	400	500	
Throughput (request/s)	13.9	23.1	26.2	31.6	–	
Error (%)	0	0	0	0	13.3	

Table 6 The test resolve token results: centralized authentication using SSO.

Resolve token (CCU)	100	200	300	400	500	
Throughput (request/s)	22.3	22.6	32.2	34.7	–	
Error (%)	0	0	0	0	9.8	

Table 7 The test request token results: decentralized authentication using DID.

Request token (CCU)	100	200	300	400	500	600	1,000	2,000	
Throughput (request/s)	93.8	96.2	136.4	185.9	196	162.6	188.8	191.6	
Error (%)	0	0	0	0	0	0	0	0	

Table 8 The test resolve token results: decentralized authentication using DID.

Resolve token (CCU)	100	200	300	400	500	600	1,000	2,000	
Throughput	11.9	41.8	61.1	103.6	107.3	109.7	138.2	–	
Error (%)	0	0	0	0	0	0	5	–	

Scenario 4: The feature test scenario of broker-less and microservice architecture

The purpose of this scenario is to check the system availability. Theoretically, BMDD is designed according to broker-less and microservice architecture, so when one service fails, other services still work. By contrast, the entire system of brokering architecture using the MQTT protocol will stop working when the MQTT broker fails. We design the test model as Fig. 27 below to test in practice.

Figure 27 Broker-less and microservice scenario test model.

We test the scenario where the user simultaneously sends data and controls the device. At the same time, we turned off the control service server (CSS) and the MQTT broker. As a result, BMDD still allows users to transfer data, but device control will be interrupted. Meanwhile, the entire system following the brokering architecture stops working when the MQTT broker is turned off. The results show that BMDD provides higher availability than brokering architectures.

Scenario 5: The feature test scenario of dynamic message transmission

The purpose of this script is to test dynamic message capabilities. As shown in the background, dynamic messaging allows users to communicate with any device, regardless of the manufacturer, because BMDD provides very high homogeneity; for example, the gateway only has two services: controlled service client (CSC) and collect data service client (CDC), whether applying BMDD to any field of IoT.

In the test mentioned in the first four scenarios, we have deployed the gateway on the Raspberry Pi module with two services: CSC and CDC, programmed by Go language. In addition, we found that gRPC can deploy on other modules, for example, ESP8266 (https://medium.com/grpc/efficient-iot-with-the-esp8266-protocol-buffers-grafana-go-and-kubernetes-a2ae214dbd29) with many other programming languages such as C/C++, python, etc. Thus, the implementation of CSC and CDC with various programming languages and hardware modules is perfectly possible.

This ability allows us to easily provide dynamic messaging when deploying BMDD. This result brings many benefits to customers, as depicted in Fig. 28. A user using the smart health service deployed on the BMDD platform in Vietnam while travelling to the US can continue to use the service without interruption even though different companies supply the devices in Vietnam and the US. This trait is one of the strong points of BMDD regarding uniformity.

Figure 28 Dynamic transmission message scenario test model.

Scenario 6: Security analysis

For the security issue, we analysis the three aspects, namely the encrypted-format packet, authentication, authorization.

In the first issue, we collect the message that is transfer among the layers. To capture the transmitted message, we use Wireshark (https://www.wireshark.org) is network packet analyzer software. Its job is to capture all network packets and then display the its content. In the encrypted message analysis, we compare the security mechanism between MQTT and gRPC protocol via wireshark tool (see Fig. 29).

Figure 29 Capture transmission message using Wireshark.

The analyzed packet’s content of MQTT and gRPC protocol is shown in Figs. 30 and 31, respectively.

Figure 30 MQTT protocol message capture.

Figure 31 gRPC protocol message capture.

According to the test result, MQTT protocol easily gets topic information as well as message content (Fig. 30); for instance the MQTT topic is “thanh-lam” and message is “Hello Thanh Lam”. Whereas, the gRPC protocol provides packet encryption (Fig. 31). This result, therefore, proves that the gRPC protocol has a better security mechanism than the MQTT protocol.

For the second aspect, we also apply Decentralized identifiers (DID) via Microsoft ION platform to design the BMDD architecture. Thanks to ION DID, BMDD identifies data objects to perform the identification separating from a third party identifier. Due to URIs, DID establishes the association between a subject with a document to build up reliable interactions, which also allowed control of the verification without testifying with a third party (see the detail in implementation section).

For the authorization, we apply the hierrachy tree to allow the user who have a full control over their data and devices. In particular, they can grant and revoke permission to access the generated data. The permission includes insert, edit, delete and so on. Moreover, they can create an unlimited level via the hierarchy tree structure. Moreover, we also apply the role-based access control in the BMDD platform. In particular, the user can manage who can access their data via the authorized permission.

Discussion

The BMDD has five contributions as mentioned in the Introduction section. Those five contributions are demonstrated through the five test scenarios mentioned in the Evaluation section.

For test scenario 1, the BMDD provides better load capacity than systems using MQTT (Nguyen et al., 2021). According to the test result (image), with the same gateway and VMs configuration, the BMDD can handle 600 CCUs without error while systems using MQTT only have a maximum of 400 CCUs. Moreover, the BMDD has a throughput two times higher than the system using MQTT.

For test scenario 2, the BMDD provides faster transmission speed and higher reliability than systems using MQTT (Thanh et al., 2021). According to the test result (image), with the same gateway and VMs configuration, the BMDD can stream 50,000 messages without error while the maximum of the system using MQTT is 400 messages.

For test scenario 3, the BMDD provides a decentralized authentication mechanism via DID ION which is based on blockchain. The decentralized authentication mechanism is suitable for the natural properties of IoT devices (mobility and distribution) (Mishra et al., 2019; Pahl & Liebald, 2019). There have been many studies on authentication mechanisms for IoT before, but mainly centralized authentication (Fremantle & Aziz, 2019; Thanh et al., 2021a).

For test scenario 4, the BMDD is designed based on broker-less and microservice architecture. This allows BMDD has no single-failure point (Kawaguchi & Bandai, 2020). In the test result, BMDD still works normally when one service is turned down while the system using MQTT depends on the central broker.

For test scenario 5, the BMDD provides dynamic message transmission mechanism that allows no distinction between data types so there is also no distinction between gateway manufacturers. For the best of our knowledge, this is a new feature, we have not found any IoT Platform that provides this capability.

Conclusion

This paper proposed the IoT platform called BMDD, a combination of broker-less and microservice architecture. These two types of architecture allow BMDD to have high availability to reduce single-point failure and scalability. Moreover, BMDD provides decentralized authentication suitable for the decentralized character of IoT devices. Furthermore, thanks to the gRPC protocol benefit, BMDD have an excellent performance as well as low power consumption. Besides, this protocol supports dynamic message transmission, which brings uniformity to IoT devices. In addition, BMDD provides logical features related to authorization, devices, and channels via decentralized identity architecture. Moreover, message queues via Kafka enhance system availability to reduce packet loss when services fail. This paper also performed the six scenarios to evaluate our above contributes in the experiment section.

For future work, we plan to improve the performance, i.e., increase and decrease the error even in the massive request token environment. The current performance can be executed in 500 CCU. For the security issues, the addition of intelligent filtering is potentially the next aim, using AI technology to limit the risk of users intentionally or being exploited to spread viruses on IoT platforms. For the traditional Kafka platform, we also plan to apply the fully homomorphic encryption (FHE) to execute the privacy-preserving computation to prevent the privacy violation behavior in the message queue. Moreover, we plan to apply the attribute-based access control (ABAC) to improve the validate process where the policy can be define in the fine-grained level.

Additional Information and Declarations

Competing Interests

Author Contributions

Data Availability

The authors declare that they have no competing interests.

Lam Tran Thanh Nguyen conceived and designed the experiments, performed the experiments, analyzed the data, performed the computation work, prepared figures and/or tables, authored or reviewed drafts of the paper, and approved the final draft.

Son Xuan Ha conceived and designed the experiments, performed the experiments, analyzed the data, prepared figures and/or tables, authored or reviewed drafts of the paper, and approved the final draft.

Trieu Hai Le performed the experiments, prepared figures and/or tables, and approved the final draft.

Huong Hoang Luong analyzed the data, authored or reviewed drafts of the paper, and approved the final draft.

Khanh Hong Vo analyzed the data, authored or reviewed drafts of the paper, and approved the final draft.

Khoi Huynh Tuan Nguyen analyzed the data, authored or reviewed drafts of the paper, and approved the final draft.

Anh The Nguyen analyzed the data, prepared figures and/or tables, and approved the final draft.

Tuan Anh Dao analyzed the data, authored or reviewed drafts of the paper, and approved the final draft.

Hy Vuong Khang Nguyen analyzed the data, authored or reviewed drafts of the paper, and approved the final draft.

The following information was supplied regarding data availability:

The data is available at GitHub:

- https://github.com/thanhlam2110/dynamic-message-server

- https://github.com/thanhlam2110/dynamic-message-client

- https://github.com/thanhlam2110/bmdd-rbac-service

- https://github.com/thanhlam2110/bmdd-collection-service

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
