# Peer review of "BMDD: a novel approach for IoT platform (broker-less and microservice architecture, decentralized identity, and dynamic transmission messages)"

_PeerJ Computer Science, doi:10.7717/peerj-cs.950_

## Round 0.1 · original submission · Major Revisions

· Academic Editor

Major Revisions

The authors propose an IoT platform called BMDD (Broker-less and Microservice architecture, Decentralized identity, and Dynamic transmission messages). The author has contributed very well in this paper, good efforts, however, some comments must be considered to make the paper in very good standard:

The introduction is extensive however it must reflect the paper idea and the proposed work.

The proposed BMDD architecture for IoT is very interesting however, there is a lack in explanation of how the BMDD will handle the data coming form things devices to the traditional network.

Security issue could have impact this proposal, how the BMDD architecture will handle this issue.

More convincing results must be provided for each scenarios with evidences.

English must be improved and proofreading is required

Extra details are needed in some parts of the literature and related work and as well in some parts of the proposal. Minimise the amount of writing.

The conclusion has not reflected the results and findings.

Reviewer 1 ·

Basic reporting

1. Abstract has not describe completely the outstanding results of this research.
2. Why used Raspberry pi, not the other microcontroller, need to justified in this paper?

Experimental design

The area of work seems to be interested. However, authors just made an experiment in lab (Amazon EC2 virtual machines (VMs) or prototype)), has no real experiment yet. We suggest authors to do more.

Validity of the findings

The results in tables 4,5,6 and 7 can be simplified again, making it easier for readers to see the comparison results.

Additional comments

no comment

Reviewer 2 ·

Basic reporting

The authors propose an IoT platform called BMDD (Broker-less and Microservice architecture, Decentralized identity, and Dynamic transmission messages) built on broker-less and microservice architecture using gRPC protocol as the primary protocol for data collection and device control. Besides, a decentralized authentication model based on blockchain technology to enhance the security of the IoT Platform will be introduced. Furthermore, BMDD also provides the function to manage user devices and channels that reduce security issues from user behavior. In addition, providing dynamic message exchange, as the purpose of this proposal, creates uniformity in IoT architecture and eliminates the distinction between different applications, data types, and IoT device manufacturers. The final recommendation is to add a message queue system to enhance platform reliability.
The paper is well structured and readable.
The paper has a good potential for being appreciated and cited, but it requires some improvements and also extension.
The section Introduction should clarify better and provide concise information with regard to the problem definition and scope of the paper.
About the related work section, each paper should clearly specify what is the proposed methodology, novelty, and results from experimentation. At the end of related works, highlight better in some lines what overall technical gaps are observed in existing works, that led to the design of the proposed approach. Innovative and self-organizing methodologies, as https://ieeexplore.ieee.org/abstract/document/9409962 should be reported.

Experimental design

What is the time complexity for the proposed algorithm?
The authors should highlight in what %age and in what parameters, the proposed methodology was found better as compared to existing ones
Analysis about scalability features of the approach could be added to further improve the strength of the paper.

Validity of the findings

The future scope of the methodology should be extended/highlighted.

---

## Round 0.2 · accepted · Accept

· Academic Editor

Accept

The paper is well structured and clear. The contribution is clear and obvious in the pare sections, the approach, implementations, good figures added to make things clear and easy to follow. The data process and figures are explained very well.